# Facilitating Access to Current, Evidence-Based Health Information for Non-English Speakers

**DOI:** 10.3390/healthcare11131932

**Published:** 2023-07-04

**Authors:** Paulo Henrique Silva Pelicioni, Antonio Michell, Paulo Cezar Rocha dos Santos, Jennifer Sarah Schulz

**Affiliations:** 1School of Health Sciences, University of New South Wales, Randwick 2031, Australia; 2Neuroscience Research Australia, University of New South Wales, Randwick 2031, Australia; 3The George Institute for Global Health, Faculty of Medicine and Health, University of New South Wales, Newtown 2042, Australia; amichell@georgeinstitute.org.au; 4Department of Computer Science and Applied Mathematics, Weizmann Institute of Science, Rehovot 7632706, Israel; paulocezarr@hotmail.com; 5The Faculty of Law and Justice, University of New South Wales, Randwick 2031, Australia; jennifer.schulz@unsw.edu.au; 6School of Population Health, University of New South Wales, Randwick 2031, Australia; 7Faculty of Health and Environmental Sciences, Auckland University of Technology, Auckland 0627, New Zealand

**Keywords:** equity, non-English communication, health information

## Abstract

Scientific communication is crucial for the development of societies and the advancement of knowledge. However, many countries, and, consequently, their researchers, clinicians and community members, lack access to this information due to the information being disseminated in English rather than their native language. In this viewpoint, we aim to discuss the impacts of this problem and also outline recommendations for facilitating non-English speakers’ access to current, evidence-based health information, thus extending the impact of science beyond academia. First, the authors discuss the barriers to accessing scientific health information for non-English speakers and highlight the negative impact of imposing English as a predominant language in academia. Next, the authors discuss the impacts of reduced access to clinical information for non-English speakers and how this reduced access impacts clinicians, clients, and health systems. Finally, the authors provide recommendations for enhancing access to scientific communication worldwide.

## 1. Introduction

Although English is not the predominant language spoken globally, and only approximately 5% of the world’s population are native English speakers [1], English is academia’s dominant language (or lingua franca). In health sciences and medicine, most journals require the submission of manuscripts in English, highlighting the dominance of English as the primary academic language for scientific communication [2]. Although “English can pose the advantage of enabling scholars to communicate with each other across borders and promote global dissemination of knowledge” [3], we recognise the inequities in science communication, which often excludes or limits access to non-English speakers. These problems faced by non-English speakers are impediments to values about access to healthcare and the goal of decolonising academia [4], which can be summed up by the phrase “leave no one behind” [5,6,7]. “Leave no one behind” is a call for action, originally contained within the United Nations and Members States’ Sustainable Development Goals, which aims to create a more equitable world where inequalities, poverty and illness are minimised and, eventually, eliminated. 

To achieve equity in science communication, academics have suggested several beneficial changes, such as abstracts in languages other than English, international boards of editors, and alternative language versions for some journals [8,9]. The Lancet editorial in 2019 also recognised the need for scientific material published in languages other than English [6]. Moreover, the Healthcare Information for All (HIFA) group was created to mitigate this need to have scientific material published in languages other than English [1,8].

However, these suggestions to address the inequities in science communication have yet to be widely and efficiently implemented. For example, with scientific globalisation, some non-English-speaking journals no longer accept manuscripts in their native language, and most health-related literature is still published in English only. While this approach is not without some justification (e.g., metrics for papers and journals increase when written in English), it leaves a range of non-English speakers behind and excludes researchers, clinicians, and customers/clients/patients. Furthermore, access to scientific information shapes the population’s understanding of public health measures and available treatments for different diseases, and this could lead to poor health outcomes in people who do not speak or do not have English as their first language. There is an urgent need to address this problem. Thus, this article aims to discuss the impact of the problem and also outline recommendations for facilitating non-English speakers’ access to current, evidence-based health information, extending the impact of science beyond academia.

## 2. Barriers to Access of Scientific Health Information for Non-English Speakers

Not all countries speak English as their native language. For example, some European countries teach young students a second language. In some European universities, the postgraduate programs are in English, making it easier for students to adopt English as the dominant language of academia [10]. However, not all countries have this advantage. Low- and middle-income countries, such as Mozambique and Brazil [7], have different educational structures, ranging from early education to post-graduate programmes. These programmes do not include the teaching of high-level English skills. Expensive fees for English schools and access problems in rural regions create barriers for non-English speakers in these countries [11,12,13]. Since non-English-speaking people from low- and middle-income countries do not have high-level English language skills, they often seek access to information using translation tools such as Google Translate. Unfortunately, the inaccuracy of Google Translate makes it particularly challenging to receive appropriate health-related information [14].

## 3. The Negative Impact of the Imposition of English as the “Universal” Language in Academia

High workloads have been recognised as a significant issue for all English and non-English speaking academics [15]. However, the authors argue that these workload issues are worse for non-English-speaking academics. Because of the language and access barriers, it takes longer for non-English-speaking academics to undertake their work. For example, non-English speakers take longer to write their grant proposals, manuscripts, and reports in a language they are not native to [13,16]. In addition, due to their lack of fluency in English, non-English-speaking researchers sometimes pay others to translate their scientific manuscripts, which creates two problems: (i) this money could have been, instead, used to purchase equipment, pay an employee, or offer scholarships [13]; and (ii) the information translated is offered and disseminated to countries where English is their first language, increasing the number and diversity of resources in these countries. This imposes two significant ethical problems too. First, there is massive investment from non-English-speaking countries in their research, which is mainly disseminated in another language. Second, the populations of non-English-speaking countries do not have access to this information due to the language barrier, and thus they do not benefit from this knowledge. For a fee, some journals offer translation services through their editors (Table 1). 

Due to the globalisation of scientific information, journals in non-English speaking countries are changing their editorial processes [17]. For example, most indexed journals in Brazil no longer accept manuscripts in Portuguese. This policy change may reflect publisher database indexing demands, whereby manuscripts must be published in English. Some journals still accept publications in English and other native languages (Table 2); however, the information is not published in both languages, limiting access to information. As mentioned above, that can magnify inequities. A corollary is that those journals, to which non-English speaker academics could access, are becoming extinct, perpetuating the inequity in science communication. This approach restricts international engagement and becomes a barrier to authorship for researchers who have the skills yet do not have English as their native language [18]. Furthermore, the metrics of journals published in non-English languages are often unfavourable because of low citations and outreach [19,20]. The unfortunate but likely outcome is that more researchers will choose journals written only in English, which, again, exacerbates the problem.

## 4. Reduced Access to Clinical Information—Impact on Clinicians, Clients, and Health Systems

Language and culture are inextricably linked when it comes to expressing our perspectives; framing our research questions; and how we engage with students, peers, patients/clients, and the public. Every time scientific information is disseminated in English alone, it increases inequity in health care. For example, most papers about COVID-19 were published in English during the pandemic [21]. At the same time, the dissemination of fake news and false information was alarming in low-to-middle-income countries, including non-English-speaking countries [22,23,24]. Moreover, for non-English speakers who rely heavily on translation tools (which are not entirely accurate), it is challenging to assess the reliability and credibility of the information. Additionally, due to the relative inaccuracy of translation tools, the interpretation, dissemination, and application of scientific English information may be biased for non-English speakers. Finally, the conceptualisation of English terms in the local language is often limited due to a restricted range of expressions that hinders clinicians, patients, and researchers from providing accurate information. 

These barriers to access occur in all health-related fields. With the current reduced number of journals that publish in non-English languages, clinicians depend on information that is reliable but expensive and not up-to-date. Researchers and clinicians worldwide write books about diverse scientific and health-related topics, and these books are often translated into non-English languages when not written by non-English speakers. However, these books usually cover a limited range of issues, sometimes missing negative or null results, often reported in systematic reviews and metanalyses, and primarily published in English. In addition, when the authors finish these books, they do not have an online update, which means that readers do not have access to the latest information [25]. The need for updated information means these clinicians are unlikely to practice evidence-based health care. For translated books, this need for updated information is even more critical. The translation process is typically restricted to a limited number of classical books and usually takes several months to publish. Thus, the information in those books is even more outdated. This, in turn, can pose risks to individuals’ health outcomes because clinicians do not have access to current scientific evidence. The unfortunate consequence is that the quality and safety of healthcare may be undermined.

## 5. Recommendations

The following is a non-exhaustive list of recommendations to begin to address the problems outlined in this article: **Non-English speakers’ task force:** We acknowledge the importance of HIFA. However, there is a need to create a task force for non-English speakers to discuss how to disseminate health-related scientific information to researchers, clinicians, and customers/clients/patients and to find other solutions.**Understand the needs of non-English speakers:** Researchers should undertake research using surveys and/or interviews to investigate how access to information for non-English speakers could be more equitable, not only for researchers but also for clinicians and community members (e.g., customers, clients, and patients). For example, this approach could give scientists data to initiate changes in editorial processes to cater to those who need access to scientific health-related information but are impeded due to language barriers.**Change in editorial handling:** Some journals now accept abstracts in languages other than English. However, the readers do not have access to the full paper. Editors and journals could allow and encourage authors who speak another language to submit their manuscripts in their native language as Appendix A. Furthermore, because of the additional work required for non-English speaking academics to prepare manuscripts, journals could “compensate” these authors for the time taken to translate the manuscript into English with a different digital object identifier (DOI). This approach would be more equitable, acknowledging the time and effort required when publishing in other languages.**Editorial “buy-in”:** Publishers like MDPI could be open to discussing the abovementioned matters. The editorial teams of journals, such as MDPI journals, could take a similar approach. When renowned publishers, editors, and journals take these approaches, it could encourage others to adopt similar processes, thereby reducing inequitable access to scientific information.

## 6. Conclusions

This article has highlighted the challenges created by having English as the dominant language of academia. As a result, there are a range of negative impacts for researchers and clinicians. Ultimately, the quality and safety of health care may be adversely impacted because non-English speaking clinicians and patients need access to the latest evidence-based information. These problems undermine our values in healthcare. For example, the World Health Organization identifies achieving universal coverage as a strategic priority focusing on quality and equity. However, this article has highlighted how the barriers that non-English speakers face in accessing scientific information hinder the fulfilment of values in healthcare. 

Clinicians, researchers, and health-policy makers face several challenges regarding translational knowledge for rural, indigenous, and/or at-risk communities. For non-English speaking countries, these challenges are amplified by the need for more access and representation in academia. In this manuscript, we emphasised scientific publication; however, English as a lingua franca also has a major impact on writing grants, conference presentations, and faculty internationalisation.

We hope that the recommendations in this article will be given thoughtful consideration and that we can fulfil our aspiration to “leave no one behind”, while building healthier societies worldwide. Versions of this manuscript are available as Appendix A in Portuguese and Spanish (Appendix A).

## Figures and Tables

**Table 1 healthcare-11-01932-t001:** The top 10 journals in the “health professions” section of the Scimago Journal Rank that offer translation services for a fee.

Journal	SJR
The Lancet Digital Health	6.433
British Journal of Sports Medicine	4.764
Qualitative Research in Sport, Exercise and Health	4.045
npj Digital Medicine	3.552
Sports Medicine	3.292
Medical Image Analysis	3.195
International Journal of Behavioral Nutrition and Physical Activity	2.709
Ultrasound in Obstetrics and Gynecology	2.572
Diabetes Technology and Therapeutics	2.374
Journals of Cardiovascular Magnetic Resonance	2.233

**Legend:** The SJR score is a value of the Scimago Journal Rank. The score is related to the weighted citations per document in each journal. The average SJR score for all journals is 1.00.

**Table 2 healthcare-11-01932-t002:** Journals in the “health professions” section of the Scimago Journal Rank part of the Scientific Electronic Library Online (SciELO), which publish papers in English and other languages.

Journal	SJR	Language
Acta Ortopédica Brasileira	0.286	Portuguese
CoDAS	0.261	Portuguese/Spanish
Revista Brasileiras de Ciências do Esporte	0.216	Portuguese/Spanish *
Hacia la Promocion de la Salud	0.178	Spanish
Revista Brasileira de Medicina do Esporte	0.177	Portuguese
Revista Cubana de informacion en Ciencias de la Salud	0.170	Portuguese/Spanish
MHSalud	0.150	Spanish
Jornal Brasileiro de Patologia e Medicina Laboratorial	0.140	Portuguese
Revista Andaluza de Medicina del Deporte	0.140	Portuguese/Spanish
Revista Cubana de Farmacia	0.116	Portuguese/Spanish
Revista Facultad Nacional de Salud Publica	0.116	Spanish

**Legend:** SJR is a value of the Scimago Journal Rank. The score is related to the weighted citations per document in each journal. The average SJR score for all journals is 1.00. * The journal publishes only in Portuguese/Spanish in certain fields of Scopus within the journal.

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
