# Peer review of "Facilitating Access to Current, Evidence-Based Health Information for Non-English Speakers"

_healthcare, 2023, doi:10.3390/healthcare11131932_

Round 1
Reviewer 1 Report
The content value is not high.
The barrier of not knowing English is not an obstacle in a situation of translation capabilities, including widely available online translations, which are of increasingly better quality.
Author Response
The citations noted throughout our paper evidence the claim that the content value of our article’s topic is very high, timely and of great interest. Our article highlights how the barriers that non-English speakers face in accessing scientific information hinder the fulfilment of values in health care and may impede quality and safe health care – such issues are of very high content value to a healthcare journal.
Not being a native English speaker remains a well-known and widely acknowledged barrier in academia; this barrier remains, despite the availability of translation services. As noted by the reviewer, these online translation tools are of “increasingly better quality”; the wording here indicates a well-known fact - online translation tools have not yet reached optimal quality, and they are most certainly not able to replicate the written prose of an experienced academic who is a Native English speaker.
Moreover, not everyone is able to access translation tools. Overall, these barriers perpetuate inequity in information, which is one of the core messages in our paper.
Reviewer 2 Report
The paper raises an old but still topical issue. Not easy to resolve as the linguistic hegemony of scientific research is strongly linked to the economic one. Now that China but also other emerging African countries are increasing their research capacity, the problem becomes even more relevant especially as regards the guarantee of health information for all. So the viewpoint turns out to be interesting and deserves publication.
If it is possible I would suggest to insert a couple of tables, one with the list of the top ten journals (with highest IF) that today offer support to researchers by accepting articles not only in English and in any case available to translate them. A second table should help us understand the scientific production in scientific journals with IF not in English language. Many countries have very good journals in their national language and the accept also papers in English. A further consideration, even if the scientific language is very essential, non-native speakers always find it difficult to carry out more refined reasoning, making full use of the most appropriate terminology of their own language. In some complex cases this can be an important limitation.
Author Response
We appreciate the reviewer’s support and comments. We created the two tables the reviewer suggested. Please see pages 3 and 4 for these tables.
Reviewer 3 Report
This is a well-written viewpoint on improving access to health information and addressing barriers for non-English speakers or people with limited English proficiency especially within the context of the imposition of English as a predominant and universal language in academia and scientific communication globally. This is an important topic that the viewpoint addresses and the recommendations and conclusion are well presented. The findings in sections 2 and 3 needs to incorporate more external research/citations.There are minor syntactical errors that can be edited with proofreading.
Author Response
We thank the reviewer for their encouraging words. As suggested, we have incorporated more external research and citations. We also undertook another revision of the entire manuscript.
Reviewer 4 Report
Thank you for the opportunity to review this manuscript. This viewpoint is interesting, timely, and important. It also seems like it has not been addressed much in academia. I like the COVID-19 example, which frames the importance of this issue.
Below are some comments that can improve the paper.
Abstract: First sentence, should it be “advancement” instead of “advance”? Next to last sentence, should impact be “impacts”?
Introduction, line 40: Can you briefly explain what “leave no one behind” is?
Recommendations: Maybe consider whether recommendations could include addressing language issues by universities, conferences, funding sources, and other professional arenas that have relevance to work published in journals? For example, there is a lot of overlap between grant proposals, papers, and conference abstracts, and journals are not the only venue that is affected by this issue. And perhaps some of the burden should be placed on universities to offer translation services for their researchers?
Lastly, just a general comment: While this is understandably focused on the negative aspect of academia’s English focus, it might be helpful to frame any positives of having a main language used in order to highlight the gaps and weaknesses of this approach. For example, perhaps in the introduction you could mention something like “Although X, Y and Z are positives of using only English in publications, these serve to highlight the challenges of this approach…”
I noted a few small grammatical errors in my comments above
Author Response
We thank the reviewer for their suggestions and encouraging words. We have fixed the abstract. The introduction also explains the “leave no one behind” concept.
Regarding the reviewer’s comments about our recommendations, we extrapolated the discussion about what bodies are equally affected by the disadvantage of not speaking English. We, however, disagree with the reviewer’s suggestion that the burden could be placed on Universities, since many of them, mainly in low-socioeconomic countries, are in precarious conditions, running on a low budget; translating papers to English would perhaps not be justifiable given the many challenges they face. Accordingly, we have not adopted that suggestion.
Round 2
Reviewer 1 Report
In my opinion, the article does not bring scientific value, the other comments coinciding with those of the first review.
Author Response
The reviewer does not define “scientific value”. We assume they mean an original scientific article that follows the standard scientific method, such as outlining aims, results, discussion and conclusions drawn from original data collection from approaches like an experiment? To use the parlance from the Healthcare journal, this type of manuscript would be an article.
By contrast, our manuscript does not purport to be an “article” nor follow the standard scientific approach because it is a viewpoint paper. For an example of a viewpoint paper published by Healthcare, please see:
https://www.mdpi.com/2227-9032/11/13/1825
Note that the purpose of a viewpoint paper is to analyse, discuss, and provoke engagement rather than to report results following the standard scientific approach.
Regarding “scientific value”, viewpoint papers may discuss issues of immense scientific value. We refer to the other three reviewers’ comments which highlight the scientific value and importance of the topic of our viewpoint manuscript. Our viewpoint manuscript engages with the scientific and other relevant scholarly literature about the topic, highlighting that the topic is of immense scientific value. It is of such scientific value that the Lancet – another authoritative scientific journal - has published a paper about the topic.
It is also important to note that for topics that intersect with other academic disciplines – such as our manuscript – authors may draw from literature outside of the sciences. For example, law and policy literature may be relevant. Those academic disciplines have their own methods, which are equally robust and relevant. It would be remiss to ignore such relevant literature simply on the basis of one reviewer who claims it merits little or no “scientific value”.